# Assessment of deep neural networks for the diagnosis of benign and malignant skin neoplasms in comparison with dermatologists: A retrospective validation study

Seung Seog Han[1]°, Ik Jun Moon[2]°, Seong Hwan Kim[3], Jung-Im Na[4], Myoung Shin Kim[5], Gyeong Hun Park[6], Ilwoo Park[7], Keewon Kim[8], Woohyung Lim[9], Ju Hee Lee[10]*, Sung Eun Chang[11]*

1 Department of Dermatology, I Dermatology Clinic, Seoul, Korea, 2 Department of Dermatology, Severance Hospital, Yonsei University College of Medicine, Seoul, Korea, 3 Department of Plastic and Reconstructive Surgery, Kangnam Sacred Hospital, Hallym University College of Medicine, Seoul, Korea, 4 Department of Dermatology, Seoul National University Bundang Hospital, Seongnam, Korea, 5 Department of Dermatology, Sanggye Paik Hospital, Inje University College of Medicine, Seoul, Korea, 6 Department of Dermatology, Dongtan Sacred Heart Hospital, Hallym University College of Medicine, Seoul, Korea, 7 Department of Radiology, Chonnam National University Medical School and Hospital, Gwangju, Korea, 8 Department of Rehabilitation Medicine, Seoul National University College of Medicine, Seoul, Korea, 9 LG Sciencepark, Seoul, Korea, 10 Department of Dermatology, Severance Hospital, Yonsei University College of Medicine, Seoul, Korea, 11 Department of Dermatology, Asan Medical Center, Ulsan University College of Medicine, Seoul, Korea

☯ These authors contributed equally to this work.
* juhee@yuhs.ac (JHL); csesnumd@gmail.com (SEC)

## Abstract

### Background

The diagnostic performance of convolutional neural networks (CNNs) for diagnosing several types of skin neoplasms has been demonstrated as comparable with that of dermatologists using clinical photography. However, the generalizability should be demonstrated using a large-scale external dataset that includes most types of skin neoplasms. In this study, the performance of a neural network algorithm was compared with that of dermatologists in both real-world practice and experimental settings.

### Methods and findings

To demonstrate generalizability, the skin cancer detection algorithm (https://rcnn.modelderm.com) developed in our previous study was used without modification. We conducted a retrospective study with all single lesion biopsied cases (43 disorders; 40,331 clinical images from 10,426 cases: 1,222 malignant cases and 9,204 benign cases); mean age (standard deviation [SD], 52.1 [18.3]; 4,701 men [45.1%]) were obtained from the Department of Dermatology, Severance Hospital in Seoul, Korea between January 1, 2008 and March 31, 2019. Using the external validation dataset, the predictions of the algorithm were

**Data Availability Statement:** The images used to test the neural networks described in this manuscript are subject to privacy regulations and

cannot be made available in totality. We provide thumbnail images of the Severance Dataset along with the demographics, clinical diagnoses, and biopsy-sites at https://doi.org//10.6084/m9.figshare.11058764. The test subset may be made available upon a reasonable request and with the approval of the IRB of the concerned university hospital. Our model in the form of an API (application programming interface) is available online at https://github.com/whria78/modelderm_rcnn_api. Contact address: whria78@gmail.com (SSH).

**Funding:** The authors received no specific funding for this work.

**Competing interests:** I have read the journal's policy and the authors of this manuscript have the following competing interests: One of the present authors (WL) is employed by LG Sciencepark. However, the company did not play any role in the study design; data collection and analysis; or the decision to prepare and submit this manuscript for publication.

**Abbreviations:** AUC, area under the curve; CNN, convolutional neural network; CT, computed tomography; IRB, institutional review board; NPV, negative predictive value; PPV, positive predictive value; ROC, receiver operating characteristic; STARD, Standards for Reporting of Diagnostic Accuracy Studies.

compared with the clinical diagnoses of 65 attending physicians who had recorded the clinical diagnoses with thorough examinations in real-world practice.

In addition, the results obtained by the algorithm for the data of randomly selected batches of 30 patients were compared with those obtained by 44 dermatologists in experimental settings; the dermatologists were only provided with multiple images of each lesion, without clinical information.

With regard to the determination of malignancy, the area under the curve (AUC) achieved by the algorithm was 0.863 (95% confidence interval [CI] 0.852–0.875), when unprocessed clinical photographs were used. The sensitivity and specificity of the algorithm at the predefined high-specificity threshold were 62.7% (95% CI 59.9–65.1) and 90.0% (95% CI 89.4–90.6), respectively. Furthermore, the sensitivity and specificity of the first clinical impression of 65 attending physicians were 70.2% and 95.6%, respectively, which were superior to those of the algorithm (McNemar test; $p < 0.0001$). The positive and negative predictive values of the algorithm were 45.4% (CI 43.7–47.3) and 94.8% (CI 94.4–95.2), respectively, whereas those of the first clinical impression were 68.1% and 96.0%, respectively.

In the reader test conducted using images corresponding to batches of 30 patients, the sensitivity and specificity of the algorithm at the predefined threshold were 66.9% (95% CI 57.7–76.0) and 87.4% (95% CI 82.5–92.2), respectively. Furthermore, the sensitivity and specificity derived from the first impression of 44 of the participants were 65.8% (95% CI 55.7–75.9) and 85.7% (95% CI 82.4–88.9), respectively, which are values comparable with those of the algorithm (Wilcoxon signed-rank test; $p = 0.607$ and 0.097).

Limitations of this study include the exclusive use of high-quality clinical photographs taken in hospitals and the lack of ethnic diversity in the study population.

## Conclusions

Our algorithm could diagnose skin tumors with nearly the same accuracy as a dermatologist when the diagnosis was performed solely with photographs. However, as a result of limited data relevancy, the performance was inferior to that of actual medical examination. To achieve more accurate predictive diagnoses, clinical information should be integrated with imaging information.

## Author summary

### Why was this study done?

- The diagnostic performance of artificial intelligence based on deep learning algorithms has been demonstrated to be superior to or at least comparable with that of dermatologists.

- However, the difference in diagnostic efficiency between algorithms and dermatologists was determined using experimental reader tests with limited clinical information related to the photographed skin abnormalities.

- Most studies performed internal validation, which indicates that both the training and validation images were selected from the same source. In addition, only a small number

of disorders have been validated in the previous studies. Thus, practical limitations and biases have complicated translation to actual practices.

### What did the researchers do and find?

- The performance of the neural network algorithm was compared with that of standard dermatologic practice for diagnosing almost all types of skin neoplasms on a large scale.

- The algorithm could successfully screen malignancy, without lesion preselection by a dermatologist. Under experimental settings, in which only images were provided for diagnosis, the performance of the algorithm was comparable with that of the 44 dermatologists who performed the reader test.

- However, the performance of the algorithm was inferior to that of the attending physicians who actually consulted with patients. This highlights the value of clinical data, in addition to visual findings, for accurate diagnosis of cutaneous neoplasms.

### What do these findings mean?

- Given photographs of abnormal skin findings, the algorithm can work ceaselessly to determine the need for dermatologic consultation at a performance level comparable with that of dermatologists.

- To further improve the algorithm's performance, metadata, such as past medical history, should be integrated with the clinical images.

## Introduction

Imaging diagnoses using deep learning algorithms [1] have shown excellent results in the analysis of fundoscopy images in ophthalmology [2], as well as X-rays [3,4] and computed tomography (CT) images [5] in radiology. In dermatology, there have been remarkable advances, leading to diagnostic performances of deep learning algorithms that are comparable with those of dermatologists for both clinical photography [6–11] and dermoscopy images [6,10,12–15]. However, it is unclear whether these excellent results can be extrapolated to the clinical setting [16].

The performance of a deep learning algorithm is dependent on the relevancy of the data exploited to train it. In particular, training an algorithm with medical data is complex, given that all clinical information cannot be collected and quantified, and the relevancy of single data domains employed for deep learning can vary substantially. For instance, even if an algorithm is trained with countless brain MRIs, we cannot be sure whether the algorithm can truly diagnose Parkinson's disease using MRI alone. Although images of skin lesions contain important and relevant information, the extent of their importance is thus far unknown for skin cancer diagnosis. Accordingly, this study was designed to compare the performance of an algorithm trained with image information alone with that of attending physicians in real clinical practice, as well as with that of physicians to whom only images were presented.

Another pitfall in implementing deep learning algorithms in a real clinical practice is the coverage of the training and test datasets. Deep learning algorithms can produce reliable results only for preselected diseases; an algorithm may demonstrate epistemic uncertainty for untrained problems [17]. Clinical questions in real practice usually involve an unlimited selection of diagnoses. The accuracy of an algorithm may be compromised if a problem encompasses a wide range of classifications or if training data do not cover appropriate conditions.

In our previous study [18], convolutional neural network (CNN) and region-based CNN were trained with 1,106,886 image crops to detect potential lesions and to predict the probability of a lesion being malignant. In that study [18], using 673 cases of 9 kinds of keratinocytic tumors, the performance of the algorithm was comparable with that of dermatologists in the experimental setting, in which diagnoses were made with multiple unprocessed images without clinical information. The aim of this study is to demonstrate the generalizability of our algorithm's performance in determining malignancy with most types of skin neoplasms and to analyze the difference in sensitivity and specificity between the experimental and real-world settings.

## Methods

### Study population

This study is reported as per the Standards for Reporting of Diagnostic Accuracy Studies (STARD) 2015 reporting guideline for diagnostic accuracy studies (S1 STARD Checklist). There was no prospective analysis plan for this study. With the approval of 2 institutional review boards (IRBs) (Severance: #2019–0571 and Asan: #2017–0087), clinical photographs were gathered from data collected at the Department of Dermatology, Severance Hospital, Seoul, Korea, retrospectively covering the period from January 1, 2008 to March 31, 2019. The information of all biopsy cases had been recorded according to a specific protocol for data collection. Clinical diagnoses, pathologic diagnosis, lesion site, biopsy site, attending physician, specimen's slide number, and information of special stains have been systematically recorded since 2005. Clinical photographs have been stored in a hierarchical structure as "root_folder/year/month/month_day/" since 2008. A total of 65 attending physicians (7.1 ± 9.5 years of experiences after board certification at the time of biopsy request; mean ± standard deviation [SD]) had recorded the clinical diagnoses on the biopsy request forms with consideration of the patient's histories and physical examinations. The IRB did not permit minors to be included in this retrospective analysis without the written consent of their parent or guardian, and it did not allow any consent requirement to be waived. Thus, all cases over 19 years of age that were pathologically diagnosed with 1 of 43 primary skin neoplasms were included (Fig 1). Pathologic diagnosis was used as a reference standard, and only cases with a single biopsied lesion were included because of possible mismatch between the lesion and diagnosis in cases with multiple biopsied lesions. Cases involving mucosal (conjunctiva and oral cavity) lesions, postoperative lesions, and laser surgery were excluded, but normal images taken for record were included, even if they did not include the lesion of interest (e.g., if photographs of both cheeks were taken, both right and left cheeks were analyzed, even if the biopsied lesion was on 1 of the cheeks only). The biopsied lesions were mostly located right in the center of photograph or indicated with either a surgical ruler or a marker. In ambiguous cases, the lesion of interest was located based on medical records including the recorded location of the lesion and its clinical and pathologic diagnosis. For grouped lesions, they were considered as a single case if they could all be considered to share the clinical and pathologic diagnoses. There were 257 (2.46%) such cases with grouped lesions. Specifically, we identified 32 such cases within the malignancy group (2 most common disorders: 13 kaposi sarcomas and 10 intraepithelial carcinomas) and 225 cases within the benign group (2 most common disorders: 70 seborrheic

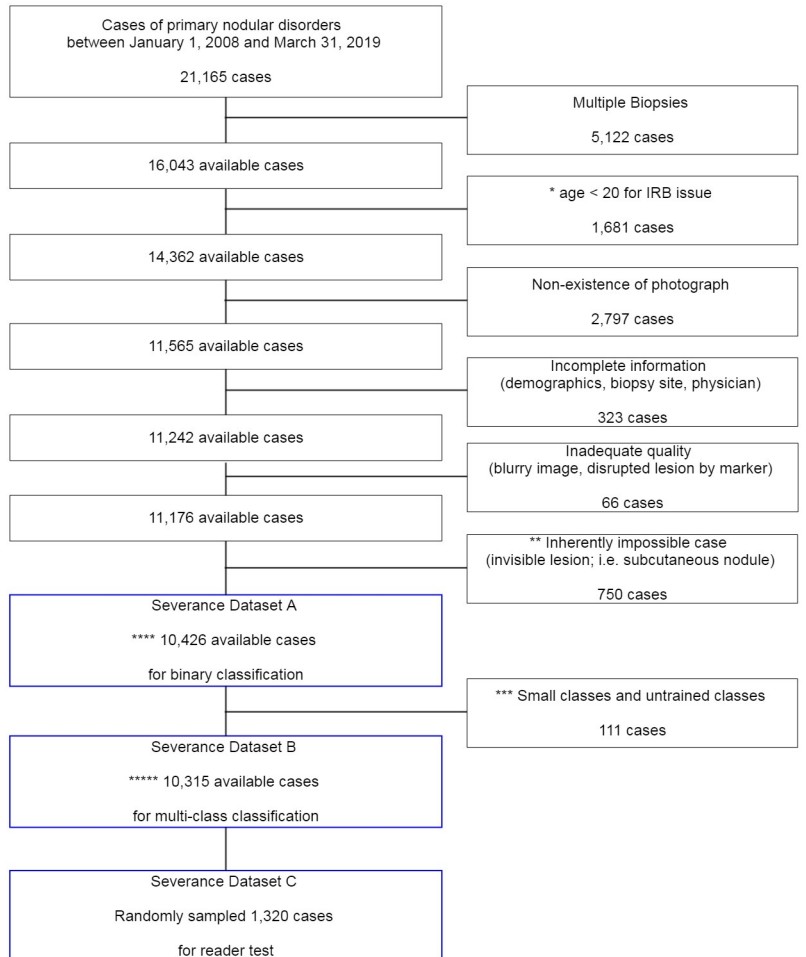

**Fig 1. Dataset selection process and exclusion criteria.** *Because the IRB did not permit enrollment of minors under the age of 20, they were excluded for analysis. **Inherently impossible cases refer to the ones where the lesion could not be exactly located based on metadata (clinical/pathologic diagnoses and record on the biopsy site). ***Small class indicates disease classes with less than 10 cases: angiofibroma, Café au lait macule, juvenile xanthogranuloma, milia, nevus spilus, and sebaceous hyperplasia. Untrained class indicates classes that the algorithm was not trained on dermatofibrosarcoma protuberans, Spitz nevus, Kaposi sarcoma, angiosarcoma, and Merkel cell carcinoma. ****Severance Dataset A: a total of 10,426 cases (40,331 images; 43 disorders; age mean ± SD = 52.1 ± 18.3, male 45.1%) used for the binary classification (cancer or not). *****Severance Dataset B: a total of 10,315 cases (39,721 images; 32 disorders; age mean ± SD = 52.1 ± 18.2, male 44.8%) used for the multiclass classification results (Fig 4, S1 and S3 Figs; S1 Table). IRB, institutional review board; SD, standard deviation.

keratoses and 44 warts). Low-quality images where the clinical features of the lesions could not be clearly identified were classified as "inadequate images." Cases where the exact location of the lesion could not be identified based on both clinical and pathological diagnoses were classified as "inherently impossible." There were 66 cases with inadequate image quality and 750 cases for which it was inherently impossible to detect lesions (Fig 1). The 3 most common disorders among the inherently impossible cases were epidermal cyst (497 cases), seborrheic keratosis (64 cases), and hemangioma (37 cases). There were 26 cases of malignant nodules (18 cases of basal cell carcinoma and 8 cases of other malignancies) among the inherently impossible cases. Finally, a total of 40,331 clinical images from 10,426 cases (9,556 patients) were included in this study (Table 1). Among the 40,331 images, a total of 35,170 images contained

**Table 1. Demographics of the Severance validation dataset.**

| | Severance Dataset | | |
| --- | --- | --- | --- |
| | **Dataset A** | **Dataset B (subset of A)** | **Dataset C (subset of B)** |
| No. of images | 40,331 | 39,721 | 5,065 |
| Patient demographics | | | |
| No. of cases (patients) | 10,426 (9,556) | 10,315 (9,464) | 1,320 (1,304) |
| Age (mean ± SD) | 52.1 ± 18.3 | 52.1 ± 18.2 | 52.0 ± 18.3 |
| Male | 4,701 (45.1%) | 4,626 (44.8%) | 580 (43.9%) |
| No. of disorders | 43 | 32 | 31 |
| Malignancy | 1,222 (11.7%) | 1,154 (11.2%) | 152 (11.5%) |
| Angiosarcoma | 17 (0.2%) | - | - |
| Basal cell carcinoma | 643 (6.3%) | 643 (6.2%) | 101 (7.7%) |
| Dermatofibrosarcoma protuberance | 9 (0.1%) | - | - |
| Intraepithelial carcinoma (SCC in situ) | 255 (2.5%) | 255 (2.5%) | 31 (2.3%) |
| Kaposi sarcoma | 39 (0.4%) | - | - |
| Keratoacanthoma | 15 (0.1%) | 15 (0.1%) | - |
| Malignant melanoma | 83 (0.8%) | 83 (0.8%) | 13 (1.0%) |
| Merkel cell carcinoma | 3 (0.0%) | - | - |
| Squamous cell carcinoma | 158 (1.5%) | 158 (1.5%) | 7 (0.5%) |
| Benign | 9,204 (88.3%) | 9,161 (88.8%) | 1,168 (88.5%) |
| Actinic keratosis | 784 (7.6%) | 784 (7.6%) | 102 (7.7%) |
| Angiofibroma | 4 (0.0%) | - | - |
| Angiokeratoma | 39 (0.4%) | 39 (0.4%) | 4 (0.3%) |
| Becker nevus | 14 (0.1%) | 14 (0.1%) | 2 (0.2%) |
| Blue nevus | 115 (1.1%) | 115 (1.1%) | 17 (1.3%) |
| Café au lait macule | 1 (0.0%) | - | - |
| Congenital nevus | 47 (0.5%) | 47 (0.5%) | 9 (0.7%) |
| Dermatofibroma | 845 (8.2%) | 845 (8.2%) | 102 (7.7%) |
| Epidermal cyst | 1,501 (14.6%) | 1,501 (14.6%) | 203 (15.4%) |
| Epidermal nevus | 26 (0.3%) | 26 (0.3%) | 5 (0.4%) |
| Hemangioma | 263 (2.6%) | 263 (2.5%) | 34 (2.6%) |
| Juvenile xanthogranuloma | 4 (0.0%) | - | - |
| Lentigo | 67 (0.7%) | 67 (0.6%) | 2 (0.2%) |
| Lymphangioma | 15 (0.1%) | 15 (0.1%) | 1 (0.1%) |
| Melanocytic nevus | 1,441 (14.0%) | 1,441 (14.0%) | 171 (13.0%) |
| Milia | 8 (0.1%) | - | - |
| Mucocele | 73 (0.7%) | 73 (0.7%) | 10 (0.8%) |
| Mucosal melanotic macule | 36 (0.4%) | 36 (0.3%) | 7 (0.5%) |
| Neurofibroma | 199 (1.9%) | 199 (1.9%) | 31 (2.3%) |
| Nevus spilus | 3 (0.0%) | - | - |
| Organoid nevus | 62 (0.6%) | 62 (0.6%) | 7 (0.5%) |
| Ota nevus | 24 (0.2%) | 24 (0.2%) | 4 (0.3%) |
| Porokeratosis | 71 (0.7%) | 71 (0.7%) | 14 (1.1%) |
| Poroma | 64 (0.6%) | 64 (0.6%) | 10 (0.8%) |
| Portwinestain | 15 (0.1%) | 15 (0.1%) | 2 (0.2%) |
| Pyogenic granuloma | 162 (1.6%) | 162 (1.6%) | 18 (1.4%) |
| Sebaceous hyperplasia | 6 (0.1%) | - | - |
| Seborrheic keratosis | 2,370 (23.1%) | 2,370 (23.0%) | 291 (22.0%) |
| Skin tag | 70 (0.7%) | 70 (0.7%) | 8 (0.6%) |

(*Continued*)

**Table 1.** (Continued)

| | Severance Dataset | | |
| --- | --- | --- | --- |
| | Dataset A | Dataset B (subset of A) | Dataset C (subset of B) |
| Spitz nevus | 17 (0.2%) | - | - |
| Syringoma | 103 (1.0%) | 103 (1.0%) | 16 (1.2%) |
| Venous lake | 101 (1.0%) | 101 (1.0%) | 13 (1.0%) |
| Wart | 636 (6.2%) | 636 (6.2%) | 83 (6.3%) |
| Xanthelasma | 18 (0.2%) | 18 (0.2%) | 2 (0.2%) |

SCC, squamous cell carcinoma; SD, standard deviation.

lesions of interest, and the remaining 5,161 images were photographs without a lesion of interest; these photographs were acquired for observation or comparison purposes. The distributions of malignant tumors by anatomical location was as follows: head and neck, 819 cases (7.9%); trunk, 162 cases (1.6%); leg, 153 cases (1.5%); and arm, 88 cases (0.8%). The distribution of benign neoplasms by anatomical location was as follows: head and neck, 4,266 cases (40.9%); trunk, 2,523 cases (24.2%); leg, 1,433 cases (13.7%); and arm, 982 cases (9.4%).

## The subsets of Severance dataset

The Severance validation dataset was obtained from the Department of Dermatology, Severance Hospital and contained 34 types of benign neoplasms and 9 types of malignant tumors.

- Severance Dataset A consisted of all the 10,426 cases (40,331 images; 43 disorders; age mean ± SD = 52.1 ± 18.3, male 45.1%).

- Severance Dataset B consisted of a total of 10,315 cases (39,721 images; 32 disorders; age mean ± SD = 52.1 ± 18.2, male 44.8%). As shown in Fig 1, the Severance Dataset B was compiled after excluding cases of belonging to small and untrained classes from the Severance Dataset A.

- Severance Dataset C consisted of a total of 1,320 cases (5,065 images; 31 disorders; age mean ± SD = 52.0 ± 18.3, male 43.9%). Severance Dataset C was composed of 1,320 randomly selected cases from Severance Dataset B. The Severance Dataset C was used for the reader test.

## Algorithm

A skin cancer detection algorithm (https://rcnn.modelderm.com) developed in a previous study [18] was used without modification. This algorithm was trained not only with images from the hospital archives but also with images that were generated by a region-based CNN [19]. The algorithm was trained with a total of 1,106,886 cropped images involving 178 disease classes. The training dataset comprised clinical images from the Asan Medical Center and images gathered from websites. The disease classifier (SENet [20] and SE-ResNeXt-50) was trained with the help of a region-based CNN (faster RCNN [19]) using a dataset that consisted of various skin lesions, as well as normal structures that could mimic pathologic lesions. A 3-stage approach was applied to efficiently build the training dataset and to maximize the explainability of the results (refer to S1 Text for more details). With the ability to detect lesions,

our algorithm can analyze clinical photographs without custom preparation that requires work by a specialist [21].

As demonstrated in Fig 2, the algorithm can detect multiple potential skin lesions in unprocessed images and report the most probable diagnoses, along with the probability in numerical values. In particular, a malignancy output was calculated from the report generated by the algorithm. We used a predefined malignancy output, as follows: malignancy output = (basal cell carcinoma output + squamous cell carcinoma output + intraepithelial carcinoma output + keratoacanthoma output + malignant melanoma output) + 0.2 × (actinic keratosis output + ulcer output) (details regarding this formula are described in S1 Text).

There are 2 different ways the algorithm may be used for lesion diagnosis: (1) the user localizes the lesion of interest (cropped-image analysis); and (2) the algorithm is used to analyze unprocessed photographs without information regarding the lesion of interest (unprocessed-image analysis). As shown in Fig 2, in unprocessed-image analysis, the blob detector and fine image selector of the algorithm detect lesions in the photograph, and the disease classifier analyzed the detected lesions and calculated malignancy outputs. The highest malignancy output among the malignancy outputs of multiple images is used as the final output score in the unprocessed-image analysis. In cropped-image analysis, the specified lesions were analyzed by the disease classifier of the algorithm, and the average malignancy output of multiple images of the lesion is used as the final output score (Fig 2). In this study, unprocessed-image analysis was employed for the binary classification of malignant and benign lesions, and cropped-image analysis was used for multiclass classification, for predicting the exact diagnosis. In the binary classification with unprocessed images, 2 cutoff thresholds (a high-sensitivity threshold and a high-specificity threshold; see definitions in S1 Text) that were defined in the previous study [18] were used to assess sensitivity, specificity, positive predictive value (PPV), and negative predictive value (NPV).

## Binary classification versus multiclass classification

Multiclass classification was used to predict the exact diagnosis, whereas binary classification was used to distinguish cancerous conditions from noncancerous ones. The binary classification was performed with Severance Dataset A, whereas the multiclass classification was performed with Severance Dataset B (Table 1). In the multiclass analysis, we excluded 6 classes (angiofibroma, Café au lait macule, juvenile xanthogranuloma, milia, nevus spilus, and sebaceous hyperplasia), where the number of cases for each class was less than 10, and we also excluded 5 classes (Spiz nevus, dermatofibrosarcoma protuberans, angiosarcoma, Kaposi sarcoma, and Merkel cell carcinoma), on which the algorithm was not trained. Consequently, the multiclass classification was performed with 32 disease classes (10,315 cases), and the binary classification was performed with 43 disease classes (10,426 cases).

## Conversion of multiclass decision to binary decision

We converted the dermatologists' multiclass decisions, namely the Top-(1)–(3) diagnoses, into binary responses using the following method: if a malignant condition was included in the Top-(n) diagnoses, the response was recorded as a malignancy. For example, provided the response "Top-1: seborrheic keratosis, Top-2: melanoma, Top-3: nevus," the Top-2 binary report was "malignancy" because a malignant condition is included in the Top-2 diagnoses. Similarly, the Top-3 binary report would also be "malignancy." This binary transformation would therefore enhance the sensitivity. On the contrary, the binary decision derived from the Top-1 response would be the most specific decision. The binary decision derived from the Top-3 diagnosis corresponds to the prediction made by the algorithm at the high-sensitivity

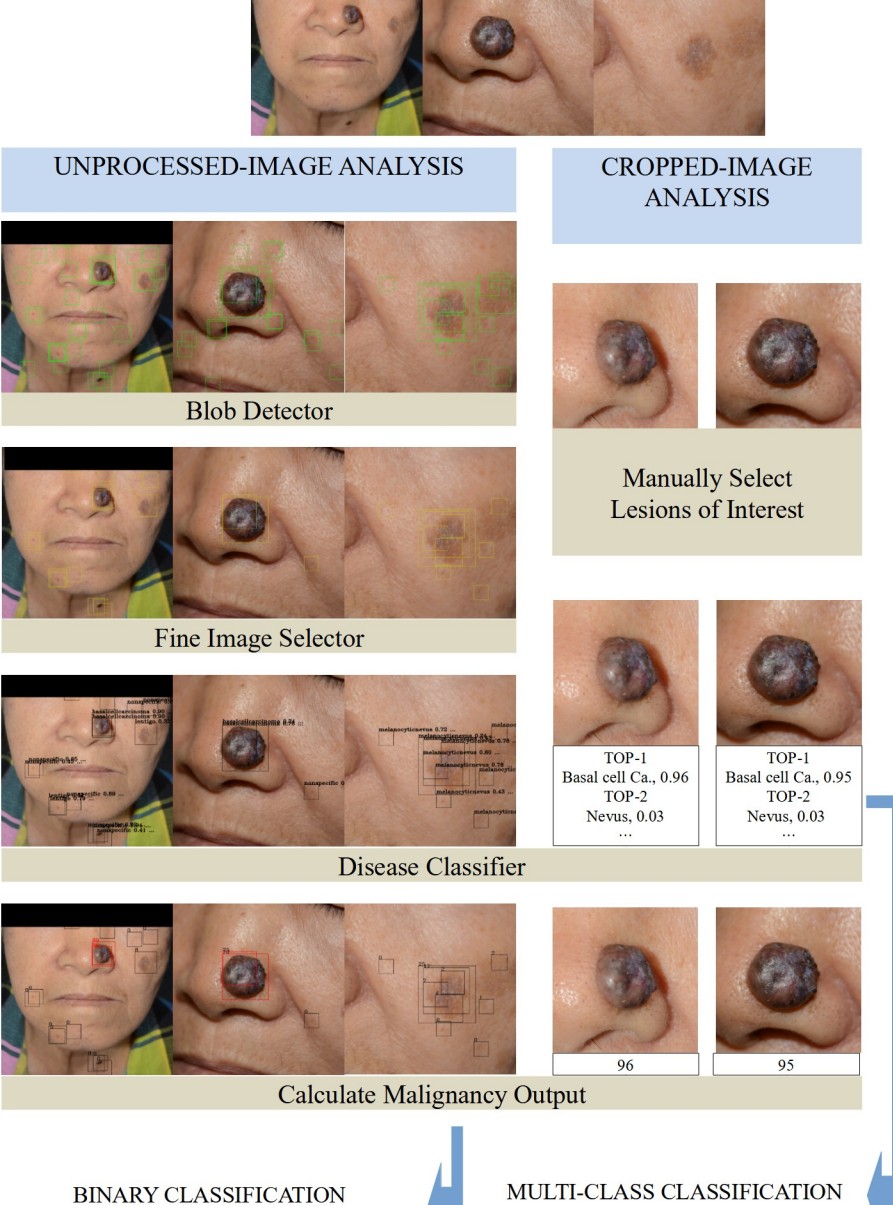

**Fig 2. Illustration of unprocessed-image analysis and cropped-image analysis by the algorithm.** F/80, pathologic diagnosis = basal cell carcinoma. In unprocessed-image analysis, (a) the blob detector looks for all possible blobs after analyzing the set of all 3 unprocessed images. (b) Then, the fine image selector marks the blobs containing possible skin lesions. (c) Finally, the disease classifier outputs the probable diagnoses and calculates the malignancy output, which is used for binary classification to determine malignancy. All boxes with a malignancy output value above the predefined high-specificity threshold (46.87) are colored in red. The highest value is included in the final report, which in this case is 90. In cropped-image analysis, (a) the user selects a lesion of interest. (b) The disease classifier reports the probable diagnoses and the malignancy output. The mean value of the 2 malignancy outputs is included in the final report, which in this case is (95 + 96) / 2 = 95.5. A multiclass classification for the exact diagnosis is carried out using the probable diagnoses put forward by the disease classifier. The final output value is the mean of the outputs from 2 CNNs as the following: Top1 = BCC 0.955, Top2 = Nevus 0.03, Top3 = . . . In this figure, the malignancy output ranges from 0 to 100, whereas the output from the disease classifier lies between 0.0 and 1.0. BCC, basal cell carcinoma; CNN, convolutional neural network.

threshold, whereas that derived from the Top-1 diagnosis corresponds to the prediction made by the algorithm at the high-specificity threshold.

### Reader test

A reader test was performed using only the photographs of lesions, blinding all clinical information. After random sampling and shuffling of image sets of 1,320 cases, 44 test sets, each containing data from 30 cases, were created (Severance Dataset C; 44 × 30 = 1,320). Each reader test consisted of different cases. A total of 44 dermatologists (5.7 ± 5.2 (mean ± SD) years of experience after board certification; in Seoul and its suburbs), where each dermatologist received a different test set, examined multiple images obtained from each patient and chose their answer from 32 choices (= 32 disorders for the multiclass classification; S1 Table).

### Test configurations

We tested 6 configurations as follows:

- Binary classification with the unprocessed images of the Severance Dataset A;

- Binary classification with the cropped images of the Severance Dataset B;

- Binary classification with the cropped images of the Severance C—Reader test;

- Multiclass classification with the cropped images of the Severance C—Reader test;

- Multiclass classification with the cropped images of the Severance Dataset B;

- Multiclass classification with the cropped images of the Edinburgh dataset.

The detailed methods and results of the multiclass classification with the Severance Dataset B and the Edinburgh dataset (10 disorders; 1,300 images; https://licensing.edinburgh-innovations.ed.ac.uk/i/software/dermofit-image-library.html) are described in S1 Text.

### Statistical analysis

With regard to the binary classification for determining malignancy, receiver operating characteristic (ROC) curves were drawn using the final output scores (the highest malignancy output in the unprocessed-image analysis and the average malignancy output in the cropped-image analysis). For the multiclass classification, to draw ROC curves of each of the classes, we used a one-versus-rest method. The area under the curve (AUC) was calculated with R (R Foundation for Statistical Computing, Vienna, Austria; version 3.4.4), using the pROC package (version 1.15.3) [22]. The confidence interval (CI) was computed using 2,000 stratified bootstrap replicates.

Top-(n) accuracy represents the fact that the correct diagnosis is among the top n predictions output by the model. For example, Top-3 accuracy means that any of the top 3 highest-probability predictions made by the model match the expected answer. Mean Top-(n) accuracies were calculated by averaging the accuracies of the 32 disorders as follows: macroaveraged mean Top-(n) accuracy = (Top-(n) accuracy of actinic keratosis + Top-(n) accuracy of angiokeratoma + . . . + Top-(n) accuracy of xanthelasma) / 32. To reflect the difference in the number of cases for each disease, the microaveraged Top-(n) accuracy was also calculated as follows: microaveraged mean Top-(n) accuracy = (Top-(n) matched cases in total) / 10,315.

Sensitivities, specificities, and Top accuracies were compared using either the McNemar or Wilcoxon signed-rank tests. DeLong test was used for the comparison between AUC values. A

95% CI was generated for all samples. A *p*-value of < 0.05 was considered to indicate a statistically significant difference.

## Results

### Binary classification of malignancy and benign lesions

With Severance Dataset A, the algorithm analyzed all unprocessed photographs of each patient to detect malignancy. For every image, 13.4 ± 16.9 (mean ± SD) lesion boxes were detected and were subject to analysis, and the AUC value was 0.863 (95% CI 0.852 to 0.875). We compared the algorithm at the predefined high-sensitivity threshold with the binary decision derived from the Top-3 clinical diagnosis. We also compared the algorithm at the predefined high-specificity threshold with the Top-1 clinical diagnosis, which produced a more specific decision. At the high-sensitivity threshold, the sensitivity and specificity were 79.1% (CI 76.9 to 81.4) and 76.9% (CI 76.1 to 77.8), respectively; in addition, at the high-specificity threshold, the sensitivity and specificity were 62.7% (CI 59.9 to 65.5) and 90.0% (CI 89.4 to 90.6) (Table 2), respectively. The sensitivity and specificity derived from the Top-3 clinical diagnoses were 88.1% and 83.8%, respectively, which were higher than those of the algorithm at the high-specificity threshold. The sensitivity and specificity derived from the Top-1 clinical diagnosis were 70.2% and 95.6%, respectively, which were also higher than those of the algorithm at the high-sensitivity threshold (Fig 3A). The difference between the results obtained for the Top-3 clinical diagnoses and the algorithm at the high-sensitivity threshold was statistically significant (McNemar test; *p* < 0.0001), and the difference between the results obtained for the Top-1 clinical diagnosis and the algorithm at the high-specificity threshold was also statistically significant (McNemar test; *p* < 0.0001).

The PPV of the algorithm was 31.3% (CI 30.3 to 32.3) at the high-sensitivity threshold and 45.4% (CI 43.7 to 47.3) at the high-specificity threshold, whereas the PPVs of the Top-3 and Top-1 clinical diagnoses were 41.9% and 68.1%, respectively. The NPV of the algorithm was 96.5% (CI 96.2 to 96.9) at the high-sensitivity threshold and 94.8% (CI 94.4 to 95.2) at the high-specificity threshold, whereas the NPVs of Top-3 and Top-1 clinical diagnoses were 98.1% and 96.0%, respectively.

### Multiclass classification for predicting exact diagnosis

The mean AUC of 32 classes obtained using Severance Dataset B (39,721 images from 10,315 cases) was 0.931 (S1 Fig and S1 Table). The macroaveraged mean Top-1/3 accuracies of the clinical diagnoses of the attending physicians were 65.4%/74.7%, and those of the algorithm were 42.6%/61.9%, respectively. The microaveraged mean Top-1/3 accuracies of the clinical diagnoses of the attending physicians were 68.2%/78.7%, and those of the algorithm were 49.2%/71.2%, respectively.

The mean AUC of 10 diseases using the Edinburgh Dataset was 0.939 (S2 Fig and S2 Table). The macroaveraged mean Top-1 and 3 accuracies were 53.0% and 77.6%, respectively.

### Reader test with board-certified dermatologists

With the Severance Dataset C (5,065 images from 1,320 cases), a reader test was conducted, where 44 dermatologists evaluated a different test set with images from 30 randomly selected cases.

For binary classification with regard to malignancy, we compared the diagnoses of the participants in the reader test with those of the algorithm. As shown in Fig 3B, the overall performance of the algorithm was comparable with those of the participants. The sensitivity of the

**Table 2. Sensitivity, specificity, PPV, and NPV of the algorithm at 2 predefined cutoff thresholds.**

| Disease | Classifier | Sensitivity (95% CI) | Specificity (95% CI) | PPV (95% CI) | NPV (95% CI) |
|---|---|---|---|---|---|
| All malignant tumors | Algorithm | | | | |
| 1,222 cases | High-sensitivity threshold | 79.1 (76.9–81.4) | 76.9 (76.1–77.8) | 31.3 (30.3–32.3) | 96.5 (96.2–96.9) |
| | High-specificity threshold | 62.7 (59.9–65.5) | 90.0 (89.4–90.6) | 45.4 (43.7–47.3) | 94.8 (94.4–95.2) |
| | Clinical diagnosis | | | | |
| | Top-1 | 70.2 | 95.6 | 68.1 | 96.0 |
| | Top-2 | 84.9 | 86.0 | 44.6 | 97.7 |
| | Top-3 | 88.1 | 83.8 | 41.9 | 98.1 |
| Basal cell carcinoma | Algorithm | | | | |
| 643 cases | High-sensitivity threshold | 81.3 (78.2–84.3) | 76.9 (76.0–77.8) | 19.8 (18.9–20.6) | 98.3 (98.0–98.6) |
| | High-specificity threshold | 66.6 (63.0–70.1) | 90.0 (89.3–90.6) | 31.7 (30.0–33.6) | 97.5 (97.2–97.7) |
| | Clinical diagnosis | | | | |
| | Top-1 | 74.0 | 95.6 | 54.2 | 98.1 |
| | Top-2 | 87.7 | 86.0 | 30.4 | 99.0 |
| | Top-3 | 90.4 | 83.8 | 28.0 | 99.2 |
| Squamous cell carcinoma | Algorithm | | | | |
| 158 cases | High-sensitivity threshold | 84.2 (78.5–89.9) | 76.9 (76.1–77.8) | 5.9 (5.5–6.3) | 99.7 (99.5–99.8) |
| | High-specificity threshold | 70.9 (63.3–77.8) | 90.0 (89.4–90.6) | 10.8 (9.7–12.0) | 99.5 (99.3–99.6) |
| | Clinical diagnosis | | | | |
| | Top-1 | 65.8 | 95.6 | 20.6 | 99.4 |
| | Top-2 | 84.2 | 86.0 | 9.3 | 99.7 |
| | Top-3 | 86.1 | 83.8 | 8.3 | 99.7 |
| Malignant melanoma | Algorithm | | | | |
| 83 cases | High-sensitivity threshold | 81.9 (73.5–90.4) | 76.9 (76.0–77.8) | 3.1 (2.8–3.4) | 99.8 (99.7–99.9) |
| | High-specificity threshold | 61.4 (50.6–72.3) | 90.0 (89.4–90.6) | 5.3 (4.3–6.1) | 99.6 (99.5–99.7) |
| | Clinical diagnosis | | | | |
| | Top-1 | 68.7 | 95.6 | 12.4 | 99.7 |
| | Top-2 | 84.3 | 86.0 | 5.1 | 99.8 |
| | Top-3 | 89.2 | 83.8 | 4.7 | 99.9 |

CI, confidence interval; NPV, negative predictive value; PPV, positive predictive value.

The algorithm analyzed unprocessed images from 1,222 cases with malignant tumors and 9,204 cases with benign neoplasms (Severance Dataset A; 43 disorders).

participants, derived from the Top-3 diagnosis, was 84.9% (CI 77.4 to 92.4). This value is slightly lower than that of the algorithm, which was 85.3% (CI 79.3 to 91.2) at the high-sensitivity threshold, but the difference was not statistically significant (Wilcoxon test; $p$ = 0.985). The sensitivity of the participants (from Top-1; 65.8%, CI 55.7 to 75.9) was also lower than that of the algorithm (66.9%, CI 57.7 to 76.0) at the high-specificity threshold, but the difference was not statistically significant (Wilcoxon test; $p$ = 0.607). The specificity of the participants (from Top-3; 66.9%, CI 62.4 to 71.4) was slightly lower than that of the algorithm (75.2%, CI 71.2 to 79.2) at the high-sensitivity threshold (Wilcoxon test; $p$ = 0.004), and the specificity of the participants (from Top-1; 85.7%, CI 82.4 to 88.9) was lower than that of the algorithm (87.4%, CI 82.5 to 92.2) at the high-specificity-threshold, but the difference was not statistically significant (Wilcoxon test; $p$ = 0.097).

For the multiclass classification for deriving the exact diagnoses, the Top-1/Top-3 accuracies of the algorithm were 49.5% (CI 46.7 to 52.4)/69.5% (CI 67.2 to 71.7), and those of the participants were 37.7% (CI 34.4 to 40.9)/53.4% (CI 49.9 to 57.0), and those in the case of the

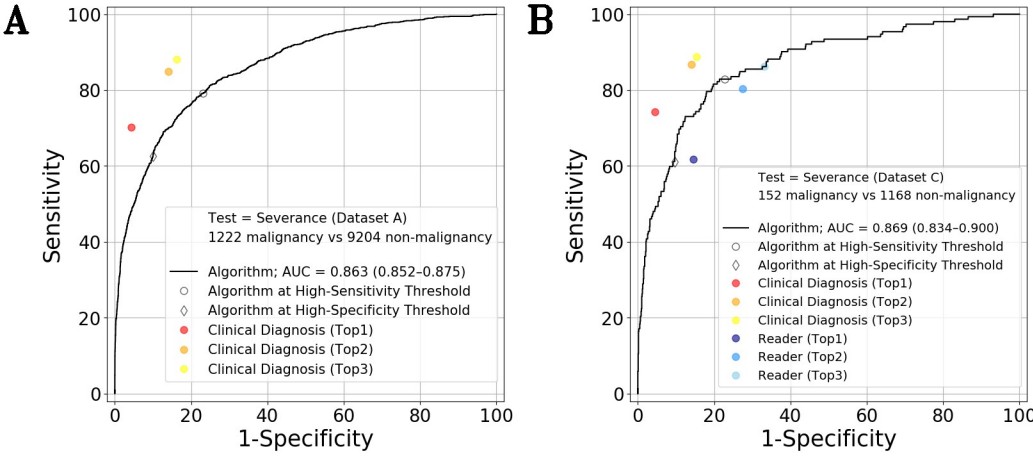

**Fig 3. Diagnostic performance for determining malignancy; binary classification.** (A) Test Dataset = Severance Dataset A (10,426 cases). (B) Test Dataset = Severance Dataset C (1,320 cases). Black curve—algorithm (unprocessed-images analysis). Diamond—algorithm at the predefined high-sensitivity threshold. Round—algorithm at the predefined high-specificity threshold. Red, orange, and yellow circles—malignancy determination derived from Top-1, Top-2, and Top-3 clinical diagnoses of the 65 attending physicians, respectively. Dark blue, blue, and sky blue circles—malignancy determination derived from Top-1, Top-2, and Top-3 diagnoses of the predictions of the 44 participants, respectively. Overall, in the binary task of determining malignancy, the performance in the case of the clinical diagnoses was superior to those of both the algorithm and the reader test participants. The performance of the algorithm and the 44 participants was comparable. AUC, area under the curve.

clinical diagnoses were 68.1% (CI 65.2 to 71.0)/77.3% (CI 74.5 to 80.2) (Fig 4). The Top accuracies of the algorithm were superior to those of the participants (Wilcoxon test; both Top 1/Top 3 accuracies; $p < 0.0001$) and inferior to those of the clinical diagnosis (Wilcoxon test; both Top 1/Top 3 accuracies; $p < 0.0001$).

## Performance changes in different settings

As shown in Fig 4, the mean Top accuracies decreased with an increase in the number of possible outputs. The algorithm was trained with 178 disease classes as a unified classifier, whereas

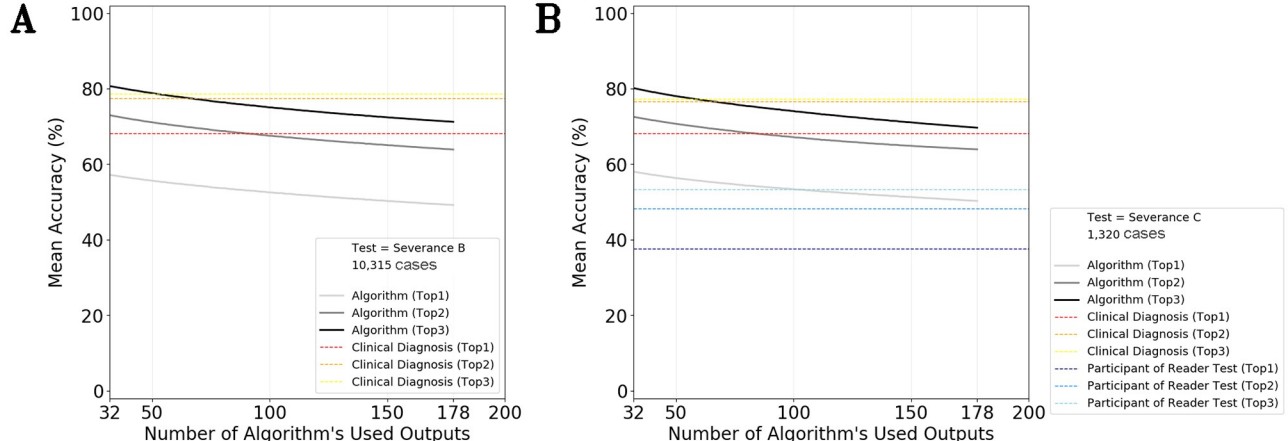

**Fig 4. Performance changes according to the number of disorders tested; multiclass classification.** (A) Test Dataset = Severance Dataset B (10,315 cases). (B) Test Dataset = Severance Dataset C (1,320 cases). Red line: 65 attending physicians. Blue line: 44 reader test participants. Black line: algorithm; among a total of 178 outputs of the algorithm, 32 to 178 outputs (x-axis) were used to calculate Top accuracies.

the test dataset included 32 classes. The prediction accuracy of the algorithm was assessed by randomly ignoring outputs outside of the 32 classes, 1 by 1, and every number between 32 and 178 was tested. Because there were many possible combinations of excluding outputs (i.e., C (156,100) = 156! / 100!(156−100)! $> 10^{43}$), the results of a bootstrap, performed 2,000 times, were plotted. When the output of the algorithm was restricted to the 32 classes, the accuracy of the algorithm (Top-3) was even higher than that of the clinical diagnosis (Top-3).

As shown in S3 Fig, the performance of the algorithm was only slightly influenced by the use of unprocessed images or cropped images. The AUC of cropped-image analysis (0.881, CI 0.870 to 0.891) was slightly better than that of unprocessed-image analysis (0.870, CI 0.858 to 0.881) (DeLong test; $p = 0.002$).

As shown in S4 Fig, the performance of the algorithm was not significantly influenced by the addition of inadequate images and images corresponding to classes that were not included during the training. The AUC values of the blue (Severance Dataset A + 750 inherently impossible cases + 66 inadequate quality cases), black (Severance Dataset A), and red (Severance Dataset A– 111 cases of small and untrained classes) curves of the S4 Fig were 0.861 (CI 0.849 to 0.872; 11,242 cases), 0.863 (CI 0.852 to 0.875; 10,426 cases), and 0.870 (CI 0.858 to 0.881; 10,315 cases), respectively. The difference between the AUCs of the black and blue curve was not statistically significant (DeLong test; $p = 0.746$), neither was the difference between the AUCs of the black and red curve (DeLong test; $p = 0.428$).

## Discussion

In this study, our algorithm showed an AUC of 0.863 in the external validation (10,426 cases; 43 disorders), which was comparable with the AUC of 0.896 in the internal validation (386 cases; 9 disorders) of the previous study [18]. The algorithm showed comparable sensitivity and specificity with those of 44 dermatologists in the experimental setting. However, the algorithm showed inferior sensitivity and specificity to those of 65 attending physicians in real-world practice.

Deep learning algorithms have recently shown remarkable performance in dermatology. In the analysis of clinical photographs [6–11] and dermoscopic images [6,10,12–15], such algorithms have shown performance comparable with that of dermatologists, with regard to the differentiation of malignant melanoma from nevus and carcinoma from benign keratotic tumors. However, direct comparisons between the results of different studies are difficult because the inclusion criteria of the validation datasets are often unclear, and the validation datasets are usually private [23]. To date, only a few studies have used external datasets to compare the performance of algorithms with that of experts [24]. In addition, there were several critiques that currently available smartphone-based applications did not reach dermatologist-level performance, as has been reported academically [25,26].

When investigating the performance of deep learning algorithms for the diagnosis of skin diseases, there are various factors that result in poor reproducibility or generalizability in actual practice.

In the algorithm training stage, these factors may include the following. (1) Limited number of training classes relative to the number of test classes: The number of disorders included in studies is usually much smaller than that encountered in real-world practice. Algorithms may well exhibit good performance in a narrow range problem, such as "melanoma–nevus binary classification." However, as shown in S1 Table, the mean Top-1 accuracy of the clinical diagnoses was only 65.4%, which suggests poor performance with regard to the initial selection of possible diagnoses. For example, an algorithm trained only for melanoma and nevus classification may not be able to diagnose nail hematoma because of epistemic uncertainty [17]. (2)

Compromised performance owing to an increase in the number of training and test classes: As shown in Fig 4, our algorithm showed better performance if the number of disorders considered was restricted to 32 disorders. However, the performance deteriorated as the number of disorders increased because there are several disorders that may appear as nodules, such as prurigo nodularis. Because we should consider more disorders than the 178 classes in real practice, the actual performance of the algorithm may be depressed to the point where the accuracy is extrapolated over 178 disease classes. Therefore, the diagnostic accuracy achieved based on a limited number of disorders may be exaggerated and not replicable in real clinical practice, where the number of conditions of concern is inexhaustible. (3) A nonrepresentative training dataset [27]: Hospital archives usually include unusual cases that require biopsies; such cases do not represent the general population. For example, there are several cases of small melanomas that do not have characteristic bizarre morphology. If an algorithm is trained on such unusual cases, the algorithm may be biased toward false positives. In addition, hospital archives have an insufficient number of images of benign disorders, which can reduce the diagnostic accuracy in the case of images from the general population with benign disorders. (4) Data leakage: This occurs when the training and test dataset are not split completely (train–test contamination) or when training uses data from the future (target leakage; e.g., cellulitis prediction based on antibiotics usage) [28]. In the case of data leakage, the model uses inessential additional information for classification [29]. (5) Specific optimization: The hyperparameters of the algorithm (i.e., batch size, learning rate, and stop iteration) may be specifically optimized for the validation dataset at the time of training and may consequently lack generalizability to other datasets [28].

In the validation stage, the factors resulting in poor reproducibility or generalizability in real practice may include the following: (1) Prediction from false hidden features: In the internal validation process, predictions may be made based on unexpected false features, instead of true features of the disease [30]. The composition of images and the presence of skin markings are likely to affect the prediction made by the algorithm [31,32]. (2) Preselection by the specialist: Preselection bias exists in real-time procedures, such as endoscopy, ultrasonography, and clinical photography. In dermatology, both training and test datasets are created by dermatologists, and dermatologists determine which lesions need to be included and which compositions are adequate. Preselection is performed by dermatologists not only with regard to the images but also with regard to the metadata. This is because metadata are selectively collected according to the physician's clinical impression. For example, dermatologists who suspect the occurrence of scabies record detailed metadata, such as the pruritus of family members, but general physicians who do not have the same suspicion may not record this information on the chart. Thus, the neural network may not exhibit the same performance without the preselection of data by dermatologists. (3) Unclear inclusion and exclusion criteria: Selection bias affects the results if cases with negative results are excluded more frequently for any reason. All patients whose data are present in the archives should be included, subject to reasonable inclusion/exclusion criteria, to prevent selection bias. (4) Data relevancy: Although algorithms may analyze images better than dermatologists, dermatologists show superior diagnostic accuracy in real practice because they consider all the patient's information for the diagnosis, whereas automated algorithms are usually trained using data with limited relevancy.

To minimize bias, this study was designed considering the following: (1) To reduce uncertainty with respect to untrained classes, most types of skin neoplasms were included. (2) To avoid selection bias, all cases biopsied in 1 hospital were included, except unavailable, inherently impossible, and inadequate-quality cases. (3) To prevent data leakage, resulting from train–test contamination, and to prove whether the algorithm was trained with true features, we tested the algorithm with 2 separate external validation datasets: the Severance dataset and

Edinburgh dataset (S1 Text). (4) To demonstrate generalizability, we verified the performance of the previously created algorithm with a new validation dataset. (5) Without dermatologist preselection, the algorithm was tested with unprocessed images.

To date, most comparisons between dermatologists and algorithms have been performed using a single cropped image of each lesion. Deep learning algorithms have shown comparable performance with that of dermatologists in such settings [6,7,9–11,13–15]. A meta-analysis of medical studies also showed that, under the same conditions, the performance of algorithms is comparable to that of healthcare specialists [24]. In this study, when the diagnosis was based on multiple cropped images without clinical information, the algorithm showed performance similar to that of the 44 readers in the binary classification test. In the multiclass classification test, the performance of the readers was inferior to that of the algorithm (Wilcoxon test; $p < 0.0001$). It was an unusual task for dermatologists to examine photographs of lesions without clinical information on the patient and to predict exact diagnosis; therefore, this unfamiliarity may have affected the multiclass results.

In this study, skin cancer diagnosis using biopsied cases from images proved to be a difficult task for experienced dermatologists, even when they were provided with all clinical information. The sensitivity calculated from 3 differential impressions of the dermatologists was 88.1%, indicating that 11.9% of malignant tumors were incorrectly classified by the dermatologists, even after thorough examination, with the presence of malignancy being demonstrated by the biopsy report. The accuracy of malignancy prediction based on the first clinical impression was 70.2%, implying a misdiagnosis rate of 29.8%. Deep learning algorithms show uncertainty regarding tasks for which they are not specifically trained [17]; therefore, if the algorithm is not a unified classifier, but a classifier trained with limited classes, the precondition may be incorrect [33].

In actual medical practice, the physician not only considers the visual information of the lesion but also various other information such as the previous medical history, referral note, and physical examination. In radiology or pathology, the reader does not only perform the diagnosis using the image alone but also checks the medical history through a chart review and reflects on clinicopathologic correlations. Currently, there exist several studies [34–36] that incorporate patients' metadata. One report showed that combining images and clinical information lead to an overall improvement in accuracy of approximately 7% [34]. In this study, the Top-1 accuracy of the clinical diagnoses (68.1%, CI 65.2 to 71.0) was substantially better than that of the 44 dermatologists (37.7%, CI 34.4 to 40.9) in the reader test with the Severance C dataset.

The strengths of this study lie in the retrospective validation on a dataset that seeks to represent the tumor distribution in an actual hospital. We used an external validation method where the external validation dataset came from sources completely different from the training data and were independent of each other.

There are several limitations to this work. Although we obtained a mean AUC of 0.939 ± 0.030 (mean ± SD; S2 Table and S2 Fig) with the Edinburgh dataset (1,300 images of 10 benign and malignant neoplasms), which primarily consists of white patients, our algorithm needs to be tested with unprocessed clinical photographs of various races and ethnicities. Given the images of dysplastic nevi, clinicians may warn of a high chance of melanoma for white patients, whereas they may recommend close observation for Asian patients, as melanoma is relatively rare in Asians [37,38]. In this study, most training and validation images acquired were of adequate quality. Image quality and composition are likely to have more of an effect on the performance of the algorithm than on human readers [31,39,40]. Therefore, the performance of the algorithm on photographs taken by the general public must be evaluated in further studies.

Standalone algorithms can play a crucial role in the screening of patients if mass diagnostic tasks are to be performed using image information alone. Regardless of the preselection of lesions, our algorithm in full automatic mode showed comparable performance to the cropped-image analysis, as shown in S3 Fig (AUC with unprocessed images = 0.870 [CI 0.858 to 0.881] versus AUC with cropped images = 0.881 [CI 0.870 to 0.891]; Test = Severance B dataset). In a previous study [18], which employed unprocessed facial images, our algorithm showed an AUC of 0.896 on the internal validation (386 patients; 81 with malignant tumors and 305 with benign neoplasms), and the performance of the algorithm was comparable with that of 13 dermatologists in terms of the F1 score and Youden index. In this study, the algorithm reproduced a similar AUC for the large-scale external validation dataset.

In this study, the sensitivity at the high-sensitivity threshold was 79.1% (CI 76.9 to 81.4); this value lies between the sensitivities (70.2% and 84.9%) derived from the Top-1 and Top-2 clinical diagnoses, as shown in Fig 3A. In a previous study, approximately 50% of the images of malignant cases were misinterpreted as benign by the general public [11,18]. We expect that algorithm-based cancer screening may facilitate appropriate referrals.

A prospective study is required to confirm the impact of algorithm-assisted improvement in diagnostic accuracy on a patient's clinical outcome. Because the algorithm's output may not always be in agreement with the user's decision, further investigation to assess how much the algorithm's output influences the user's clinical practice is needed. Further, frequent use of a diagnostic algorithm portends the risk of false positive outputs. Thus, research on its optimal use should be conducted in the near future.

## Conclusions

Because of the limited data relevancy and diversity involved in differential diagnoses in practice, the performance of the algorithm was inferior to that of dermatologists. However, in the experimental setting, the performance of the algorithm was comparable with that of the participants, which was consistent with existing reports [24]. Without the preselection of lesions by dermatologists, our standalone algorithm showed an AUC of 0.863 (Severance A dataset), which demonstrates its unprecedented potential as a mass screening tool. More clinical information, such as patient metadata, may be incorporated to further improve the performance of the algorithm.

## Supporting information

**S1 STARD Checklist. Standards for Reporting of Diagnostic Accuracy (STARD 2015).**
(DOCX)

**S1 Text. Supplementary methods and results.**
(DOCX)

**S1 Fig. Multiclass task—ROC curve of the algorithm with 32 skin tumors in the Severance B dataset.** DER_Top1 –Top-1 accuracy of the clinical diagnoses. DER_Top2 –Top-2 accuracy of the clinical diagnoses. DER_Top3 –Top-3 accuracy of the clinical diagnoses. The algorithm analyzed multiple cropped images from the Severance Dataset B (39,721 images of 10,315 cases; 32 disorders). Not 32 outputs, but all 178 outputs were used for analysis without restriction.
(TIF)

**S2 Fig. Multiclass task—ROC curve of the algorithm for 10 skin tumors in the Edinburgh dataset.** The algorithm analyzed 1,300 images from the Edinburgh dataset (https://licensing. edinburgh-innovations.ed.ac.uk/i/software/dermofit-image-library.html). All images in the

Edinburgh dataset were cropped around the lesion of interest. All 178 outputs were used for analysis without restriction. We drew the ROC curves in a one-versus-rest manner.
(TIF)

**S3 Fig. Performance comparison between unprocessed-image analysis and cropped-images analysis.** Test = Severance Dataset B (10,315 cases). Black curve—algorithm (unprocessed-image analysis). Gray curve—algorithm (cropped-image analysis). In the unprocessed-image analysis, the algorithm detected lesions of interest from multiple clinical photographs, and the algorithm predicted whether the detected lesions were malignant or not. In the cropped-image analysis, the algorithm predicted whether multiple lesions of interest that were chosen by the user were malignant or not. The AUC of 0.881 (0.870 to 0.891) obtained for the cropped-image analysis was slightly better than the AUC of 0.870 (0.858 to 0.881) obtained for the unprocessed-image analysis (DeLong test; $p = 0.002$).
(TIF)

**S4 Fig. Performance comparison with the subsets of the Severance Dataset.** Black curve (10,426 cases); Test = Severance Dataset A. Blue curve (11,242 cases); Test = Severance Dataset A + 750 inherently impossible cases + 66 inadequate quality cases. Red curve (10,315 cases); Test = Severance Dataset A– 111 cases of small and untrained classes = Severance Dataset B. The algorithm was tested in the unprocessed-image analysis mode when images were added and deleted based on Severance Dataset A. We compared the binary performance of the algorithm with regard to malignancy detection using multiple subsets of the Severance dataset. The AUC values of the blue, black, and red curve were 0.861 (11,242 cases), 0.863 (10,426 cases), and 0.870 (10,315 cases), respectively. The difference between the black and blue curve was not statistically significant (DeLong test; $p = 0.746$), and neither was that between the black and red curves (DeLong test; $p = 0.428$).
(TIF)

**S1 Table. Multiclass task—AUCs and Top accuracies of the algorithm compared with those of clinical diagnoses for the 32 skin tumors of the Severance B Dataset.** The algorithm analyzed multiple cropped images from Severance Dataset B (39,721 images of 10,315 cases; 32 disorders). We calculated the AUC values of the ROC curves in a one-versus-rest manner.
(DOCX)

**S2 Table. Multiclass task—AUCs and Top accuracies of the algorithm for 10 skin tumors in the Edinburgh Dataset.** The algorithm analyzed 1,300 images from the Edinburgh dataset (https://licensing.edinburgh-innovations.ed.ac.uk/i/software/dermofit-image-library.html). All images in the Edinburgh dataset were cropped images around the lesion of interest. We calculated the AUC values of the ROC curve in a one-versus-rest manner.
(DOCX)

## Acknowledgments

We would like to thank the professors and clinicians who participated in the reader test. Thanks to Lee Joongyub, MD of Seoul National University for his advice on statistics. Thanks to Park Jeoongsung, PhD of Qualcomm for his help in stabilizing the web-DEMO.

## Author Contributions

**Conceptualization:** Seung Seog Han, Ik Jun Moon, Seong Hwan Kim, Jung-Im Na, Myoung Shin Kim, Ilwoo Park, Keewon Kim, Woohyung Lim, Sung Eun Chang.

**Data curation:** Ik Jun Moon, Seong Hwan Kim, Jung-Im Na, Myoung Shin Kim, Gyeong Hun Park, Ju Hee Lee, Sung Eun Chang.

**Formal analysis:** Seung Seog Han, Ik Jun Moon, Jung-Im Na, Gyeong Hun Park, Ilwoo Park.

**Investigation:** Seung Seog Han, Ik Jun Moon, Seong Hwan Kim, Jung-Im Na, Myoung Shin Kim, Gyeong Hun Park, Ilwoo Park, Keewon Kim.

**Methodology:** Seung Seog Han, Ik Jun Moon, Seong Hwan Kim, Jung-Im Na, Gyeong Hun Park, Ilwoo Park, Keewon Kim, Woohyung Lim.

**Project administration:** Seung Seog Han, Ju Hee Lee, Sung Eun Chang.

**Resources:** Ik Jun Moon, Seong Hwan Kim, Jung-Im Na, Myoung Shin Kim, Gyeong Hun Park, Ju Hee Lee, Sung Eun Chang.

**Software:** Seung Seog Han, Woohyung Lim.

**Supervision:** Ju Hee Lee, Sung Eun Chang.

**Validation:** Seung Seog Han, Ik Jun Moon, Gyeong Hun Park, Ilwoo Park, Keewon Kim.

**Visualization:** Seung Seog Han, Ik Jun Moon, Keewon Kim.

**Writing – original draft:** Seung Seog Han, Ik Jun Moon, Keewon Kim.

**Writing – review & editing:** Seung Seog Han, Ik Jun Moon, Jung-Im Na, Ilwoo Park, Keewon Kim, Sung Eun Chang.

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
