## [Editor Report · Decision Letter 0]

15 Jan 2020

Dear Dr Chang, 

Thank you for submitting your manuscript entitled "Retrospective Assessment of Deep Neural Networks for 43 Skin Tumor Diagnosis in Comparison with Dermatologists in Real Practice" for consideration by PLOS Medicine.

Your manuscript has now been evaluated by the PLOS Medicine editorial staff [as well as by an academic editor with relevant expertise] and I am writing to let you know that we would like to send your submission out for external peer review.

**Please be aware that, due to the voluntary nature of our reviewers and academic editors, manuscript assessment may be subject to delays during the holiday season. Thank you for your patience.**

Kind regards,

Adya Misra, PhD,

Senior Editor

PLOS Medicine

---

## [Decision Letter · Decision Letter 1]

20 May 2020

Dear Dr. Chang,

Thank you very much for submitting your manuscript "Retrospective Assessment of Deep Neural Networks for 43 Skin Tumor Diagnosis in Comparison with Dermatologists in Real Practice" (PMEDICINE-D-20-00108R1) for consideration at PLOS Medicine. 

[LINK]

In light of these reviews, I am afraid that we will not be able to accept the manuscript for publication in the journal in its current form, but we would like to consider a revised version that addresses the reviewers' and editors' comments. Obviously we cannot make any decision about publication until we have seen the revised manuscript and your response, and we plan to seek re-review by one or more of the reviewers. 

We expect to receive your revised manuscript by Jun 10 2020 11:59PM. Please email us (plosmedicine@plos.org) if you have any questions or concerns.

We look forward to receiving your revised manuscript. 

Sincerely,

Emma Veitch, PhD

PLOS Medicine

On behalf of Clare Stone, PhD, Acting Chief Editor,

PLOS Medicine

plosmedicine.org

*We'd suggest revising the title to fit with PLOS Medicine's usual style as well as to ensure the importance and content of the article is clearer to a broad, multidisciplinary audience. Normally the title would begin with main concept/study question then the study design in the subtitle (ie, after a colon). eg, "Assessment of Deep Neural Networks for Skin Lesion Diagnosis in Comparison with Dermatologists: Retrospective Validation Study".

*As noted by one reviewer, much of what is included in some of the figure legends probably belongs within the Methods section of the main text, and similarly some of the supplementary appendix may also need to be considered to move into the main text (and as main text figures) where that material is key to understanding the core methods and findings of the study.

*At this stage, we ask that you include a short, non-technical Author Summary of your research to make findings accessible to a wide audience that includes both scientists and non-scientists. The Author Summary should immediately follow the Abstract in your revised manuscript. This text is subject to editorial change and should be distinct from the scientific abstract. Please see our author guidelines for more information: https://journals.plos.org/plosmedicine/s/revising-your-manuscript#loc-author-summary

*Did your study have a prospective protocol or analysis plan? Please state this (either way) early in the Methods section.

*In places, the English writeup for the paper could be improved (as there are a few non-grammatical phrasings) and therefore we'd suggest asking a native English speaking colleague or contact, if you can, to help proofread the revised version before resubmitting it. As examples, the following:

*Introduction line 83: "Imaging diagnoses using deep learning algorithm" - "Imaging diagnoses using deep learning algorithmS..."

*Introduction line 92: "However, clinical problems are more complex and relevancy of a single domain data employed for deep learning can quite vary.." >this does not flow/make sense very well and could be clarified. How about "However, clinical problems are more complex, and THE RELEVANCE of SINGLE DATA DOMAINS employed for deep learning can vary SUBSTANTIALLY" 

*Introduction line 95: "images of the skin contain important relevant information, the amount of their importance is yet unknown in skin cancer diagnosis" > "images of the skin contain important AND relevant information, BUT the EXTENT OF THEIR importance is THUS FAR unknown FOR skin cancer diagnosis"

(These are just some quick examples, there may be more throughout the text so we'd recommend getting further input before resubmitting). 

Comments from the reviewers:

Reviewer #1: The authors provide an important report that sheds light on the current state of image based diagnostic AI for skin cancer, highlighting the limitations of reader studies that omit clinical metadata. Use of an external dataset for validation is a significant strength of the study.

The definition of 'malignancy output' is intriguing, and deserves additional explanation, perhaps in a supplementary table that specifically address the reasoning for inclusion of actinic keratoses (with a 0.2x modifier) and the inclusion of 'ulcer' as malignant.

On a technical level: Soft-max outputs are not themselves designed to be subjected to thresholded analysis. They are relative scores, in relation to other outputs. Was the softmax layer removed before thresholding? This could significantly change results. Authors should also explain why did they choose to study multiclass for only cropped images? Would suggest inclusion of multiclass performance on both cropped and non-cropped images, as well as a malignancy score. Lastly, more details might be included in the paper that highlight the quantifiable differences between the training and the test datasets.

Reviewer #2: "Retrospective Assessment of Deep Neural Networks for 43 Skin Tumor Diagnosis in Comparison with Dermatologists in Real Practice" validates a previously-described machine learning algorithm (cited as [18], published as "Keratinocytic Skin Cancer Detection on the Face Using Region-Based Convolutional Neural Network", JAMA Dermatology, 2019) by mostly the same authors, on validation data obtained from Korea's Severance Hospital from 2008 to 2019. As discussed in the paper, this serves to reduce certain biases (e.g. selection bias, data leakage, generalizability, preselection) in evaluating the real-world performance of the algorithm. Additional supplementary performance on a publicly-available Edinburgh dataset of Caucasian subjects was also reported.

While dermatologist-level classification of skin lesions has been claimed since 2017 [citation 6], it remains true that many practical limitations and biases (some covered by the authors in the discussion) have complicated translation to actual practices. The particular strengths of this study lie in the retrospective validation on a dataset that seeks to represent the tumor distribution in an actual hospital. The main findings that automated diagnosis based on images only might not match up to standard clinical diagnosis, could be considered as valuable insight into the complications encountered in translating research systems to practice. That said, there may be a number of issues with the manuscript that warrant attention:

1. In the selection of the main study population (Severance Datasets), it is stated that "Only cases with a single lesion were included because of possible mismatch between the lesion and diagnosis in patients with multiple lesions" (Line 116). However, this would appear to undercut one of the main functionalities of the algorithm being validated, which is described in [18] as being able to detect multiple candidate lesions from unprocessed clinical images via R-CNN. Moreover, this further suggests that the study population might not be as unbiased as later promoted, since it has been restricted to patients with only a single lesion, beforehand.

In particular, was this filtering of patients with multiple lesions, done before the initial step of the selection flowchart (Fig 1), since "multiple lesions" does not appear to be a listed exclusion step? The authors might disclose the number of patients with multiple lesions if so, and if possible explain further why testing on multiple lesions was not plausible (e.g. no lesion-level annotations?)

2. It is stated that "...normal images taken for record were included, even if they did not include the lesion of interest". Does this refer to misclassified diagnoses (i.e. some lesion is listed in the records, but the corresponding image does not show that lesion), or is there some other interpretation? In either case, the number of such cases might also be listed if possible.

3. In general, there do not appear to be any representative validation dataset lesion images for either the unprocessed or the cropped analyses, unlike in say [18] and many other previous papers. The authors might consider including some such example images.

4. For the binary classification to determine malignancy experiment, it is stated that the "receiver operating characteristic (ROC) curves were drawn using the final output scores (the highest malignancy output in the unprocessed-image analysis and the average malignancy output in the cropped-image analysis)" (Line 156). However, it is not clear how these scores (from the unprocessed and cropped analyses) were combined. This might be clarified.

5. The derivation of malignancy output (Line 131) might be discussed. In particular, how were the weights defined (0.2 for actinic keratosis/ulcer, 1.0 for other outputs)? Was there some objective function or impact considered in the formulation of this equation?

6. The clinical diagnoses of the 65 attending physicians used for the binary classification experiment, might be further described. In particular, what protocol/clinical data was used/available? What is the experience level of the physicians if available (as stated for the readers), and were the images graded by multiple physicians with arbitration, or by single physicians only?

7. In Line 177, the Fig 1 label does not appear to reflect the Top-1 clinical diagnoses discussed.

8. For binary classification of malignancy vs. benign, other than Top-1 and Top-3, a natural measure might be to aggregate the predicted probability of all malignant classes, against the predicted probability of all benign classes, in diagnosing malignancy. Additionally, it is not clear how binary classification would involve Top-k metrics, since the objective would appear to be to diagnose malignancy vs. benign, and not the specific tumor type.

Moreover, the relevant of a Top-k metric towards actual practical use is not very clear. For instance, given a malignant tumor, if the algorithm predicts a benign tumor as most probable, and the correct malignant tumor as second most probable, this is a true positive under Top-2; however, would this be referred by the algorithm for further attention? Conversely, if the algorithm predicts incorrect malignant tumor types for the three most probable diagnoses, this is a false negative, but the tumor would correctly be referred.

9. For the presented main results (Figure 2), how was the gold standard diagnosis (against which both the algorithm and the physicians/readers were compared against) obtained?

10. For the accuracy against number of algorithm's used outputs graphs (Figure 3), the set of number of outputs for which actual performance was obtained might be provided (i.e. was every number between 32 and 178 tested, and if not, which numbers were tested and the remainder interpolated?)

11. While the algorithm being validated has been described in a previous publication [18], the authors might consider more fully covering its details (e.g. training parameters, how the various component CNNs were combined) in the supplementary appendix.

12. Much of the information presented as Figure descriptions/captions might be more appropriately located in the main text.

13. A number of minor grammatical/spelling issues might be addressed, e.g. "were higher 88.1% / 83.8% of" (Line 176), "hospital achieves usually include" (Line 231), "skin cancer diagnosis is proved to be a difficult task" (Line 274); overall, the authors might consider having the manuscript proofread further.

Reviewer #3: The authors validate a CNN for the diagnosis of skin tumors in comparison with dermatologists on an external validation set of 40,331 images. It is a large study that has merit and depth but needs some improvement. 

1. The expression "skin tumor" in the title and in the main text of the manuscript is misleading because it suggests that the target diseases are only malignant neoplasms. In colloquial language skin tumor means skin cancer. The title suggests that the test set included 43 types of skin cancer. The categories included in the test set include benign neoplasm, malignant neoplasms, hamartomas, and cysts. Lentigo, for example, is not a tumor, it is flat. Cysts are not tumors. The generic term tumor should be avoided. The title should be something like "43 classes of solitary skin lesions" or "43 classes (categories) of common, non-inflammatory solitary skin lesions", but certainly not skin tumors. 

2. Data availability (training set, test set, and code) is very restrictive and is probably not in line with the open data policy of the PLOS journal family. Authors should rethink their decision. Also the code of the CNN should be made available to let others repeat the experiments. 

3. The results section needs some revision. Outcome measures with confidence intervals should be reported instead of p-values only. 

4. In the results section the authors wrote: As shown in Fig 2B, overall, the sensitivities and specificities of the algorithm were comparable with those of the participants (Wilcoxon test; P=0.99/0.0043 for sensitivity/specificity of the Top-3 of the participants versus sensitivity/specificity of the algorithm at the high-sensitivity threshold; P=0.61/0.097 for sensitivity/specificity of the Top-1 of the participants versus sensitivity/specificity of the 196 algorithm at the high-specificity threshold). 

It is very difficult to understand this sentence and to extract the correct numbers for the outcome measures sensitivity and specificity. It is inappropriate to give only P-values without outcome measures. An easier metric would be to report the number of correct diagnosis by batch. 

5. The authors further wrote: We calculated mean accuracy by averaging the accuracies of the 32 disorders as follows: macro-averaged mean Top-(n) Accuracy = (Top-(n) Accuracy of actinic keratosis + Top-(n) Accuracy of angiokeratoma + … + Top-(n) Accuracy of xanthelasma) / 32.

This fits better to the methods section and is also difficult to understand. I am not sure but could it be that the authors refer to the average recall or the mean sensitivity? It needs clarification.

6. It is also unclear if the ± refers to confidence intervals or standard deviations. 

7. When the authors speak of accuracy in the multiclass problem do they mean frequency of correct diagnoses? Please clarify

8. It is unclear how the 1,320 patients were selected from the master set to create the batches for the reader study. 

9. Much more information is need from the 44 dermatologists who participated in the reader study and how they were selected. Is there a risk of self selection bias, recruitment bias etc.

[LINK]

---

## [Decision Letter · Decision Letter 2]

7 Jul 2020

Dear Dr. Chang,

Thank you very much for submitting your manuscript "Assessment of Deep Neural Networks for the Diagnosis of Benign and Malignant Skin Neoplasms in Comparison with Dermatologists: Retrospective Validation Study" (PMEDICINE-D-20-00108R2) for consideration at PLOS Medicine. 

[LINK]

In light of these reviews, I am afraid that we will not be able to accept the manuscript for publication in the journal in its current form, but we would like to consider a revised version that addresses the reviewers' and editors' comments. Obviously we cannot make any decision about publication until we have seen the revised manuscript and your response, and we plan to seek re-review by one or more of the reviewers. 

We expect to receive your revised manuscript by Jul 21 2020 11:59PM. Please email us (plosmedicine@plos.org) if you have any questions or concerns.

We look forward to receiving your revised manuscript. 

Sincerely,

Adya Misra, PhD

Senior Editor 

PLOS Medicine

plosmedicine.org

Abstract

Please add further background here to provide context for this study and help readers understand the importance of this work

Please add further information about the datasets used and the clinicians who participated. It would be helpful to distinguish between 65 attending physicians and 44 dermatologists to note their specific role in this study. 

The last sentence of the methods and findings section must include 2-3 limitations of your work

Conclusions

Please temper these, especially “a mass screening tool in a telemedicine setting” as this was not directly tested.

Please ensure that the study is reported according to the STARD 2015 reporting guideline for diagnostic accuracy studies, and include the completed STARD checklist as Supporting Information. Please add the following statement, or similar, to the Methods: "This study is reported as per the STARD 2015 reporting guideline for diagnostic accuracy studies (S1 Checklist)."

The STARD guideline can be found here: http://www.equator-network.org/reporting-guidelines/stard/

Author Summary

Lines 86-95 should be simplified as it is meant to be a non technical summary. Please replace “convolution neural networks” and “lesions” with less technical language to convey that the aim of this work is to compare the diagnostic efficiency of an algorithm versus a dermatologist.

Line 98- please avoid assertions of primacy

Line 113-quality of images is not mentioned previously in this section and I would suggest this is removed or further context provided in earlier bullet points

Introduction

Please add a space between text and reference brackets throughout the text. Please note all full stops should be after square brackets. 

Please clearly state the aim of your work towards the end of the introduction. Please also briefly provide context of the algorithm and any previous work describing it. 

Please consider if this sentence is relevant "the complete sequence and location of white and black stones are required to play game Go; thus, DeepMind’s AlphaGo[17] was trained with this data to surpass human champions". We recommend removing this. 

Methods

Did your study have a prospective protocol or analysis plan? Please state this (either way) early in the Methods section. Please explicitly state that none of these analyses were pre-planned, as requested in the previous version. 

Please provide additional information regarding where this study was carried out (city,country), where were the patient images obtained from and where were the dermatologists based. 

Can you briefly provide the information about severance data sets A ,B,C in the main text? 

Results

Please provide exact p values where appropriate, except when p<0.001. Please also ensure all p values are provided up to three decimal places

Discussion

Please present and organize the Discussion as follows: a short, clear summary of the article's findings; what the study adds to existing research and where and why the results may differ from previous research; strengths and limitations of the study; implications and next steps for research, clinical practice, and/or public policy; one-paragraph conclusion.

Lines 351-353 require revision for clarity and to avoid the double use of “studies” 

Line 372 should be revised “ do not represent the general population”

In the limitation section you mention an Edinburgh dataset which has not been mentioned before. Please provide the necessary details in the methods and results section

Please add citations to support this statement “as melanoma is relatively rare in

466 Asians”. 

Please do not use the term Caucasians and use “white” instead. 

Data statement- please can you clarify if the algorithm is available to interested researchers, providing a link for access whether free or requested here. For the test subset, please provide a contact person for data requests. Please note that in both instances, authors cannot act as sole sources of the data. 

Bibliography must be in Vancouver style

Most of the information in figure legends is not provided in the methods section. This was specifically requested in the previous version of the manuscript. Specifically, the information about various datasets has not been included in the methods section and how unprocessed image analysis versus cropped image analysis is carried out is also not in the methods. 

Is the information on page 27 lines 578 onwards intended to be a figure legend? I am not sure this is appropriate, please place this information in the results section. The same goes for text on Page 30 which should be in the methods section only. I note that you have added lines 166-169 but it needs to be in the same level of detail as page 30. 

Comments from the reviewers:

Reviewer #2: We thank the authors for addressing most of our concerns from the previous review round. A few points might be however further considered:

1. In the response to Point 1, it is stated that "photographs of biopsied lesions also contain many other different skin lesions that are located in proximity". In such cases, how is the actual targeted biopsied lesion identified (e.g. is its position specified in a separate data/annotation file, or is it assumed to be obvious from the given label)? This might be described in the text.

2. Reference is made to "inherently impossible" cases. Exactly why these cases are impossible might be briefly explained (e.g. label not present in training set, etc)

3. The response for Point 8 was helpful in understanding the details of the algorithm assessment, and the authors might consider including a possibly-abridged version in the supplementary material.

4. In line 385/409, does "innocent bias" have some widely-accepted technical meaning? If not, it might be termed as "hindsight bias" or "leakage bias", which appear to be more commonly used.

5. In Line 422, it might be clarified "In this study, skin cancer diagnosis using biopsied cases [from images] proved to be a difficult task...", if our interpretation is correct here.

6. In Line 462, "Caucasians subjects" -> "Caucasian subjects"

7. In Line 474, "of dermatologist" -> "of dermatologists"

8. For the Edinburgh evaluation, it might be explicitly stated whether the model output was restricted to the 10 classes.

9. The figures appear to exhibit image artefacts in the manuscript PDF, although the original TIF/TIFF images appear alright.

Reviewer #3: While I am still disappointed about their restrictive data sharing policy, the authors substantially improved the manuscript in the revised version.

[LINK]

---

## [Editor Report · Decision Letter 3]

7 Aug 2020

Dear Dr. Chang,

Thank you very much for re-submitting your manuscript "Assessment of Deep Neural Networks for the Diagnosis of Benign and Malignant Skin Neoplasms in Comparison with Dermatologists: Retrospective Validation Study" (PMEDICINE-D-20-00108R3) for review by PLOS Medicine.

I have discussed the paper with my colleagues and the academic editor and it was also seen again by xxx reviewers. I am pleased to say that provided the remaining editorial and production issues are dealt with we are planning to accept the paper for publication in the journal.

[LINK]

We look forward to receiving the revised manuscript by Aug 14 2020 11:59PM. 

Sincerely,

Adya Misra, PhD

Senior Editor 

PLOS Medicine

plosmedicine.org

Requests from Editors:

Title : please add “a” in the title descriptor to change to “a retrospective validation study”

Abstract line 92- instead of “predictions” do you mean to say “predictive diagnoses” perhaps? 

Please place full stops after the reference brackets

Please copyedit the author summary section for English language and usage. Please also add bullet points to the summary

Figure 1: please revise the box “age <20 for IRB issue” to clearly indicate what the issue is

As stated in previous decision letters, it is important for the methods section to state whether analyses were planned or not, regardless of whether the study was prospective or retrospective. Please state in the methods that there was no prospective analysis plan for this study, as you have indicated in the previous response to our request. 

(https://rcnn.modelderm.com) – as mentioned in a couple of places as developed in their previous study – the link is broken. Please provide a working link

Line 489 – “To prevent data leakage” – please clarify?

Line 556 – “We expect that algorithm-based cancer screening will facilitate appropriate referrals.” - instead, use ...the algorithm may…

Line 551 – “The algorithm can operate endlessly with minimal cost” should be removed as there is no cost analysis in this study

Line 572 – “patient metadata, should be exploited “ perhaps exploit isn’t the wisest of words here and 'should' can go also in place of 'may'

Please remove the filled consent form for privacy reasons and include a blank consent form for publication in the original language and an English translation. 

STARD checklist- please use sections and paragraphs instead of page numbers as these will change during publication 

Please submit your algorithm and API in a public repository such as GitHub or Sourceforge or similar. In line with PLOS data policies, we ask that all methods, software used in studies such as these are provided without restriction. Please provide the URL and or accession number within the data statement. The restrictions on clinical images are acceptable. 

Comments from Reviewers:

[LINK]

---

## [Editor Report · Decision Letter 4]

19 Oct 2020

Dear Dr Chang, 

On behalf of my colleagues and the academic editor, Dr. Harald Kittler, I am delighted to inform you that your manuscript entitled "Assessment of Deep Neural Networks for the Diagnosis of Benign and Malignant Skin Neoplasms in Comparison with Dermatologists: A Retrospective Validation Study" (PMEDICINE-D-20-00108R4) has been accepted for publication in PLOS Medicine. 

PRODUCTION PROCESS

Before publication you will see the copyedited word document (within 5 business days) and a PDF proof shortly after that. The copyeditor will be in touch shortly before sending you the copyedited Word document. We will make some revisions at copyediting stage to conform to our general style, and for clarification. When you receive this version you should check and revise it very carefully, including figures, tables, references, and supporting information, because corrections at the next stage (proofs) will be strictly limited to (1) errors in author names or affiliations, (2) errors of scientific fact that would cause misunderstandings to readers, and (3) printer's (introduced) errors. Please return the copyedited file within 2 business days in order to ensure timely delivery of the PDF proof. 

If you are likely to be away when either this document or the proof is sent, please ensure we have contact information of a second person, as we will need you to respond quickly at each point. Given the disruptions resulting from the ongoing COVID-19 pandemic, there may be delays in the production process. We apologise in advance for any inconvenience caused and will do our best to minimize impact as far as possible.

PRESS

PROFILE INFORMATION

Thank you again for submitting the manuscript to PLOS Medicine. We look forward to publishing it. 

Best wishes, 

Adya Misra, PhD

Senior Editor 

PLOS Medicine

plosmedicine.org